# Expression plasticity regulates intraspecific variation in the acclimatization potential of a reef-building coral

Crawford Drury [1] ✉, Jenna Dilworth[1,2], Eva Majerová[1], Carlo Caruso[1] & Justin B. Greer [3]

Phenotypic plasticity is an important ecological and evolutionary response for organisms experiencing environmental change, but the ubiquity of this capacity within coral species and across symbiont communities is unknown. We exposed ten genotypes of the reef-building coral *Montipora capitata* with divergent symbiont communities to four thermal pre-exposure profiles and quantified gene expression before stress testing 4 months later. Here we show two pre-exposure profiles significantly enhance thermal tolerance despite broadly different expression patterns and substantial variation in acclimatization potential based on coral genotype. There was no relationship between a genotype's basal thermal sensitivity and ability to acquire heat tolerance, including in corals harboring naturally tolerant symbionts, which illustrates the potential for additive improvements in coral response to climate change. These results represent durable improvements from short-term stress hardening of reef-building corals and substantial cryptic complexity in the capacity for plasticity.

Traditional Hawaiian culture supports a reciprocal connection with coral reefs[1], including those in Kāneʻohe Bay, Oʻahu, Hawaiʻi where this research is based. We hope to honor this relationship by recognizing its foundational importance in this work.

Climate change presents a fundamental challenge to organisms, which must adjust to environmental shifts via plasticity or adaptation[2]. Phenotypic plasticity, the ability of a genotype to exhibit different phenotypes in divergent environments[3], is particularly important in the context of rapid climate shifts because it occurs within a generation. This short-term response to rapid spatial or temporal change may bridge the gap to reproductive events during which evolution can occur[4], potentially allowing for adaptation and genetic rescue. Acclimatization, a form of plasticity which maintains fitness in different conditions, is particularly important for ectotherms[5] and sessile organisms such as reef-building corals, which live near their upper thermal limits[6].

Coral reefs are declining worldwide due to a combination of stressors[7], with climate change impacts such as increasing sea surface temperatures and ocean acidification posing a major threat to their long-term persistence[8]. Exposure to increased sea surface temperatures disrupts the symbiotic partnership between the coral host and dinoflagellate algae in the family Symbiodiniaceae[9] in a process termed coral bleaching[10]. The frequency and severity of coral bleaching events has increased in recent decades[11] and under current emissions scenarios bleaching is expected to occur annually on most reefs by mid-century[12].

These rapidly changing environmental conditions underscore the key role that phenotypic plasticity plays as a mechanism through which sedentary organisms can respond to climate change[13]. Intra-generational plasticity in the coral response to thermal stress can arise from shifts in algal symbiont community[14], microbiome composition[15] and host gene expression patterns[16], allowing corals to exhibit

[1]Hawaiʻi Institute of Marine Biology, Kāneʻohe, HI, USA. [2]University of Southern California, Los Angeles, CA, USA. [3]U.S. Geological Survey, Western Fisheries Research Center, Seattle, WA, USA. ✉e-mail: crawford.drury@gmail.com

plasticity in growth, morphology, skeletal characteristics, and bleaching tolerance[17–19]. Marine invertebrates exposed to environmental stress also exhibit a range of epigenetic marks, including methylation, which impact gene expression and phenotype[20–22] and are impacted by symbiont dynamics[23].

Changes in gene expression are a crucial mechanism for phenotypic plasticity, serving as a link between genetic and cellular processes and physiological responses to environmental stressors[16]. Upon exposure to thermal stress, corals mount a dynamic transcriptomic response involving genes related to the regulation of heat shock proteins, apoptosis pathways, ion transport, and metabolism[24–26]. Distinct gene expression signatures differentiate the response to acute and sustained thermal stress[27]; while certain components of the transcriptome may be upregulated as soon as 1 h after exposure[28,29], long-term stress results in broad-scale downregulation of metabolic processes[30]. Gene expression changes also occur in response to sublethal thermal stress, allowing corals to acclimatize to increased temperatures and resulting in improved performance under subsequent re-exposure[31–34].

Plasticity, stress-hardening, or induced acclimatization have also been identified as promising mechanisms for the conservation of coral reefs[35,36] and are an emerging area of research. A key unresolved question is how the baseline level of a trait influences the magnitude of flexibility of a coral colony, which is particularly important given the complexity of interactions between the host animal, symbionts, and the microbiome. Previous work has documented both negative and positive relationships between these factors[37–40], representing the presence of an ultimate 'ceiling' for acclimatization or its absence, respectively.

Our understanding of the duration of change created by expression plasticity and its ability to support acclimatization is also limited by experimental approaches which have primarily evaluated phenotype immediately after pre-exposure[32,33,41]. However, a few experiments have used moderate recovery periods on the order of 2 weeks and found that persistent gene expression changes explained improved thermal tolerance[29,34], particularly in apoptosis and cell-death pathways. The relative importance of temperature variability is also disputed; regimes with more variation can confer higher thermal tolerance on corals than constant pre-exposures[42,43], but this effect is not universal[32,33,44].

Here, we apply genome-wide gene expression profiling to investigate the response of ten unique *Montipora capitata* genotypes with differential bleaching phenotypes to four pre-exposure treatments

before evaluating thermal performance in a stress test. *M. capitata* is a dominant reef-building coral in Kāneʻohe Bay, Oʻahu, Hawaiʻi[45] and hosts algal symbionts of both *Cladocopium* and *Durusdinium*[46,47]. During a bleaching event in 2015, paired colonies that displayed differential bleaching phenotypes (bleached vs non-bleached) were tagged, resulting in an in situ library of corals with known bleaching history[46,48,49]. Generally, these bleaching phenotypes are associated with differential algal symbiont communities[50], following the canonical heat tolerance differences between *Cladocopium* and *Durusdinium*. Here we show that genotype, historical phenotype, and pre-exposure treatment have a significant effect on performance during stress testing, where two treatments created a durable positive effect. We also document no relationship between baseline thermal tolerance and the magnitude of plasticity and a strong correlation between one gene and the capacity for acclimatization.

## Results

### Stress-testing

We exposed ten genotypes of *M. capitata* to four different high temperature pre-exposure profiles for 5 days before returning them to their collection site (Fig. 1). We stress tested corals 124 days after the conclusion of pre-exposure treatments, at which point there was no difference in initial photosynthetic efficiency between phenotypes (two-way ANOVA $F = 0.303$, $p = 0.582$). There was a significant difference between treatments (two-way ANOVA $F = 2.498$, $p = 0.042$), which may represent a latent effect of pre-exposures; however, this outcome was due to a single comparison (pulse increase vs pulse high, Tukey HSD $p = 0.037$). Historically nonbleached corals experienced a 10% decline in photosynthetic efficiency significantly more slowly (2.3 eDHW, Wilcox $p = 0.007$) than historically bleached corals (Fig. 2a) recapitulating previously observed patterns of higher thermal tolerance in *Durusdinium*-dominated genotypes compared to *Cladocopium*-dominated *M. capitata*[46,50,51] and stable symbiont community structure in both bleaching phenotypes during thermal stress[46,51].

In both phenotypes, corals from the control treatment experienced a 10% decline in *fv/fm* significantly more quickly than corals from the constant high treatment (Fig. 2b; bleached $p = 0.038$, nonbleached $p = 0.003$) and the pulse treatment (Fig. 2b; bleached $p = 0.033$, nonbleached $p = 0.029$). There were no significant differences in bleaching rate between control and the pulse increase or pulse high pre-exposures for either phenotype ($p > 0.05$), although these ED10 values were higher in all cases (Fig. 2b).

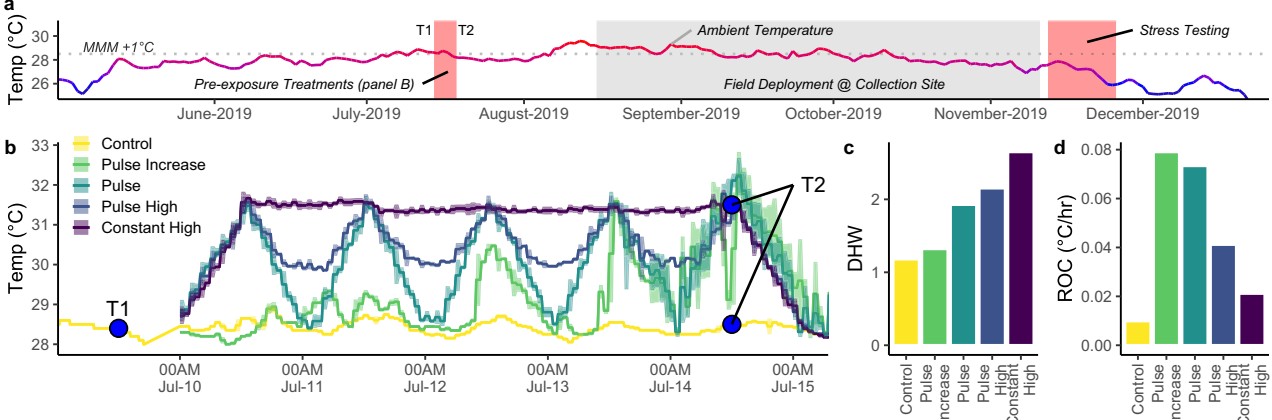

**Fig. 1 | Experimental Design. a** Experimental timeline showing pre-exposure treatments, field deployment and stress testing, overlaid on ambient temperatures. **b** Five day pre-exposure treatments including a control treatment at ambient temperature and four elevated profiles with different characteristics. **c** Experimental Degree Heating Weeks during pre-exposure treatments. Note that

even controls accumulated ~1DHW due to the ambient conditions (reference **a**). **d** Rate of change during pre-exposure treatments, calculated as °C/hr, averaged across replicate tanks. Treatment colors are maintained across figures. Source data are provided as a Source Data file.

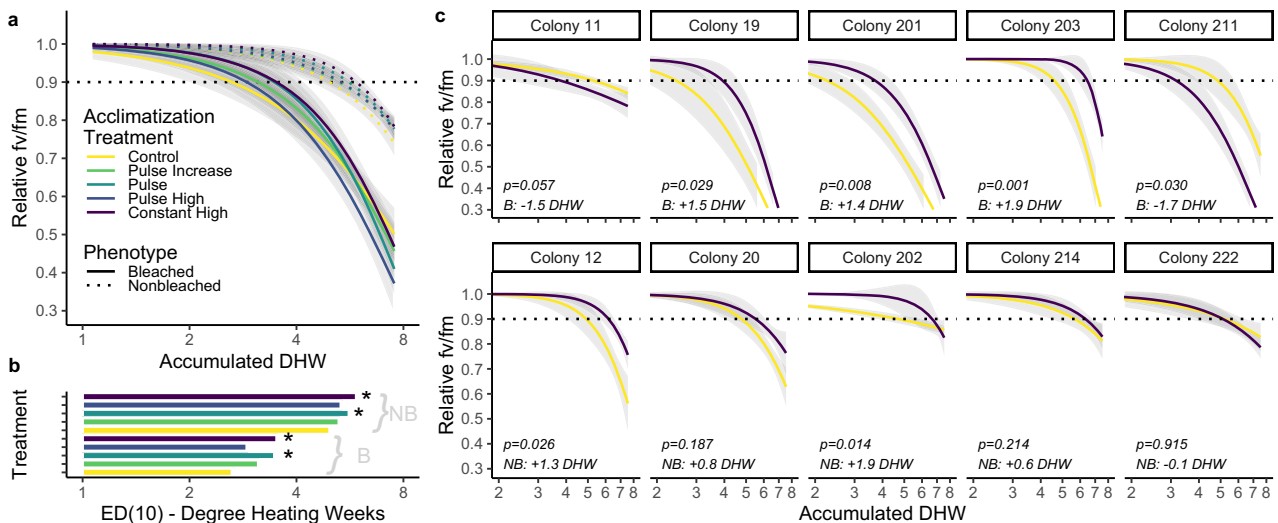

**Fig. 2 | Stress-testing Responses. a** Overall stress testing responses for all fragments ($n = 299$) 124 days after pre-exposure treatment. Colors correspond to Fig. 1 and line type denotes historical phenotype. Dotted horizontal line represents 10% decline in relative fv/fm. **b** Mean ED10 estimates for each phenotype in each treatment, where larger values represent slower bleaching during thermal stress ($n = 30$ per treatment × phenotype). Shared x-axis with (**a**). Asterisks represent significant differences in ED10 in a treatment compared to control in a two-way ANOVA (NB: CH-C $p = 0.003$, P-C $p = 0.028$; **b**: CH-C $p = 0.038$, P-C $p = 0.033$).

**c** Genotype-specific response to thermal stress in corals from the control and constant high pre-exposure treatments. Colors and dotted line correspond to (**a**). Historically bleached corals are in the top row, historically nonbleached corals are in the bottom row, with uncorrected two-way t-test $p$ values comparing treatments and DHW gained or lost (Constant High minus Control) noted in each panel. Colors correspond to Fig. 1 and represent treatment. All fits include shaded error intervals which represent the 95% CI generated by *drc*. Source data are provided as a Source Data file.

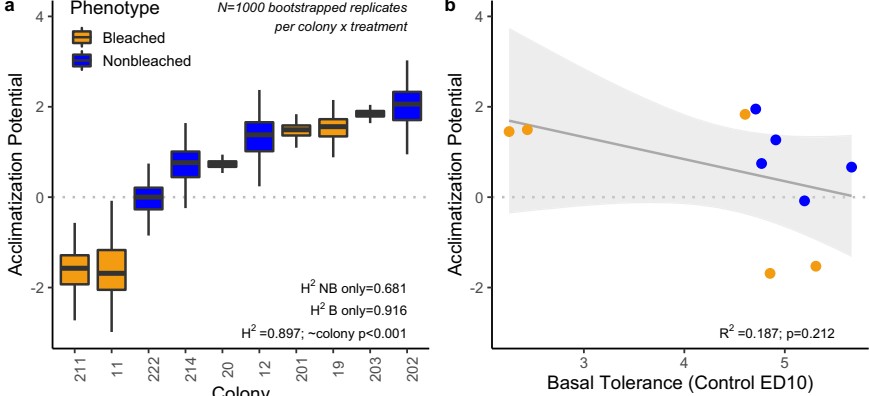

**Fig. 3 | Broad-Sense Heritability of Acclimatization does not relate to Basal Performance. a** Bootstrapped acclimatization potential of each genotype showing high heritability of acclimatization potential ($n = 1000$ per genotype). Boxplots include first and third quartiles, median line and whiskers at ±1.5 interquartile range, compared using a one-way ANOVA. **b** Basal thermal tolerance was quantified for each genotype as the ED10 during stress testing in corals from the control pre-

exposure (i.e., no previous heat treatment) and compared with the acclimatization potential, defined as the difference in DHW between constant high and control ED10 using a linear regression. Points are colored by phenotype; the gray line represents overall relationship and shaded area is 95% CI of linear regression fit. There was no significant relationship for either phenotype when analyzed separately. Source data are provided as a Source Data file.

## Genotypic effects in acclimatization

The constant high treatment elicited the largest positive response and there was substantial genotypic variation in acclimatization potential, including both positive and negative changes (Fig. 2c). Corals pre-exposed to the constant high treatment gained or lost between −1.7 and 1.9 DHW during the subsequent heat stress when compared to controls. Seven of ten genotypes experienced a net positive effect, and the overall impact was $0.61 ± 0.41$ DHW (mean ± se; one-sample t-test $\mu = 0$; $p = 0.086$). One genotype had a significant negative effect due to constant high pre-exposure ($p < 0.05$), while 5 genotypes had a significant positive effect ($p < 0.05$). There was no difference in the magnitude of impact between phenotypes (t-test $p = 0.512$). Interestingly, colony 11, which has an atypical symbiont community (for its

assigned binary phenotype) and hosts moderate amounts of *Durusdinium*, experienced moderate declines more similar to the historically nonbleached corals, which are dominated by *Durusdinium*.

The bootstrapped estimate of broad-sense heritability ($H^2$) of acclimatization potential was 0.897 for all corals (Fig. 3a) and acclimatization potential was significantly different between genotypes (one-way ANOVA, $p < 0.001$). Broad-sense heritability estimates were nearly 35% higher when analyzing only historically bleached corals ($H^2 = 0.918$) than when analyzing only historically nonbleached corals ($H^2 = 0.681$). There was little explanatory power and no significant relationship between basal thermal tolerance and acclimatization potential (Fig. 3b; $R^2 = 0.187$, $p = 0.212$). This relationship was not significant when examining only bleached ($p = 0.192$) or nonbleached

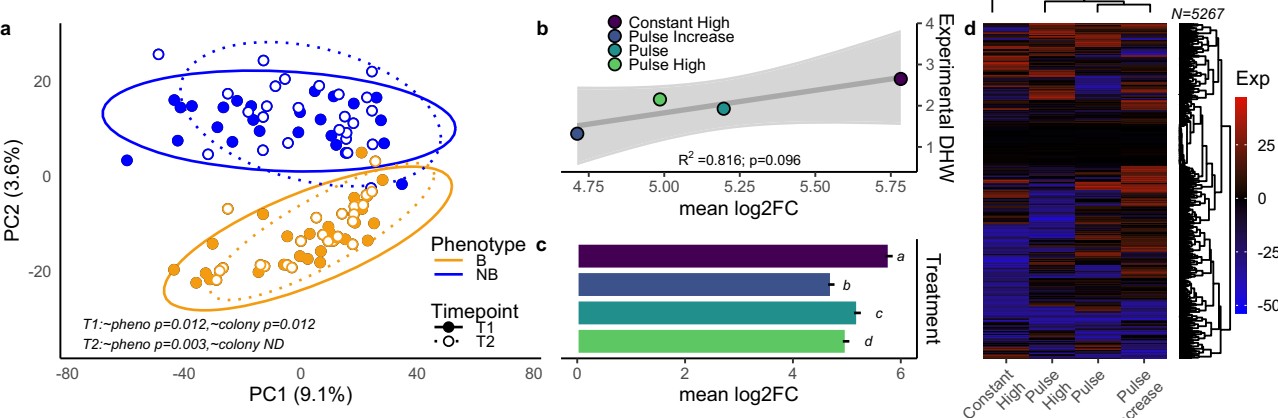

**Fig. 4 | Gene Expression Dynamics. a** Global gene expression patterns before and after pre-exposure for each phenotype. Color represents phenotype and line type represents timepoint. Note design did not include replication of colony at T2, so no analysis is performed. Timepoints and phenotypes were compared using PERMANOVA. **b** Relationship between experimental Degree Heating Weeks during pre-exposure and average $\log_2$Fold-Change of all contigs. The gray line represents overall relationship and shaded area is 95% CI of linear regression fit. Color represents treatment. **c** Mean absolute value of $\log_2$Fold-Change for all contigs in each treatment ($n = 17,826$ per treatment). Letters designate significant differences from Tukey's HSD post-hoc testing ($p < 0.05$) after a one-way ANOVA and $x$-axis corresponds to (**b**). Error bars are standard error. Color represents treatment (**d**) Heatmap of all contigs which were differentially expressed in any treatment ($n = 5287$), with hierarchical clustering of treatments and contigs. Color represents expression value. Source data are provided as a Source Data file.

corals ($p = 0.335$) and was not significant when removing two large negative outliers (Fig. 3b; $p = 0.334$).

### Transcriptomic response to pre-exposures

Global transcriptomic patterns were significantly impacted by phenotype at both timepoints (Fig. 4a; $p = 0.012$ for each) and by colony at T1 ($p = 0.012$). Treatment had no significant effect on global patterns of gene expression at T2 (8.4% variance explained, $p = 0.218$), but significantly impacted a total of 5267 contigs across the four treatments (Fig. 4d). Total accumulated heat stress during treatments (eDHW, Fig. 1c) explained 81.6% of the variation in mean $\log_2$ fold-change across all contigs (Fig. 4b), which was significantly different between each treatment and highest in the constant high acclimatization exposure (Fig. 4c).

The large number of genes uniquely differentially expressed in the constant high treatment (Supplemental Fig. 2) and the unique expression profile indicated by hierarchical clustering (Fig. 4d) suggest that this treatment elicited a significantly different response. We calculated $z$-scores differentiating response in constant high from all other treatments and found gene ontologies relating to ribosomes, peptide synthesis and metabolism and reaction to oxygen-containing compounds were significantly enriched in contigs with high $z$-scores (Supplemental Table 1), indicating functional differentiation in the response to the constant high treatment. Among these ontologies, 8 of 11 were also significantly upregulated in a meta-analysis of the generalized stress response in *Acropora*[27], including peptide metabolism and biosynthesis, cellular responses to endogenous stimulus and structural constituents of ribosomes.

Despite the distinct molecular responses of each pre-exposure, both the constant high and pulse treatments elicited a shared significant response in 15% of genes that were significant in at least one treatment. These genes, which may represent core molecular responses to thermal challenge, included many of the ontologies in a generalized response to severe stress[27], including the activation of an innate immune response, positive regulation of protein ubiquination and apoptosis and downregulation of replication.

### Expression-acclimatization correlates

The high heritability of acclimatization potential suggests genotype-specific correlate, so we calculated the control-corrected response to the constant high pre-exposure for each contig in each genotype. We then correlated this value with that genotype's acclimatization potential, the gain or loss of thermal tolerance during stress testing to examine gene regulation corresponding to phenotypic shifts (Fig. 5a). Of 12,772 contigs passing filters, the largest and most significant correlation was −0.965 (FDR $p = 0.073$) for Cluster 77005 (Fig. 5b). Using GO_MWU, we examined ontologies enriched in large correlation values and found 16 significant functions (FDR $p < 0.1$), including associations with ATPase activity, ubiquitin-like protein ligase binding, lipid catabolism and ribosomes (Fig. 5c). Blastn analysis of Cluster 77005 returned uncharacterized LOC122956333 (XP-044171934.1) from *Acropora millepora* as the closest match, and amino acid searches via blastx identified similar uncharacterized proteins from other aquatic species as significant hits. We then performed functional analysis on the full length *Acropora millepora* amino acid sequence and identified a conserved AAA + ATPase domain (Supplemental Table 2) or Rho GTPase domain, both suggesting nucleotide-binding proteolytic hydrolase activity and supporting the enrichment of ATPase-related functions in the overall GO analysis.

## Discussion

The persistence of sensitive ecosystems under climate change requires both short- and long-term responses to stress. For sessile ectotherms that have limited ability to move to more hospitable conditions, intragenerational plasticity is particularly important, especially under the rapid shifts expected under climate change. Here we show that shifts in gene expression during pre-exposure are associated with a latent, durable improvement in thermal tolerance in a reef-building coral. This acclimatization effect is genotype-specific, highly heritable and unrelated to basal thermal tolerance, highlighting substantial scope for plasticity across symbiont communities and individual coral colonies.

Our results show a striking expansion of the known duration of acclimatization effects in corals, using an acute pre-exposure which still elicits an ecologically relevant change after nearly 4 months. We did not collect RNA samples during the recovery period and there is little information available about the persistence of expression changes over time after the removal of stress; however, corals experiencing natural bleaching undergo transcriptomic shifts can which persist for 6 months to a year[52,53]. The durability of this effect suggests that

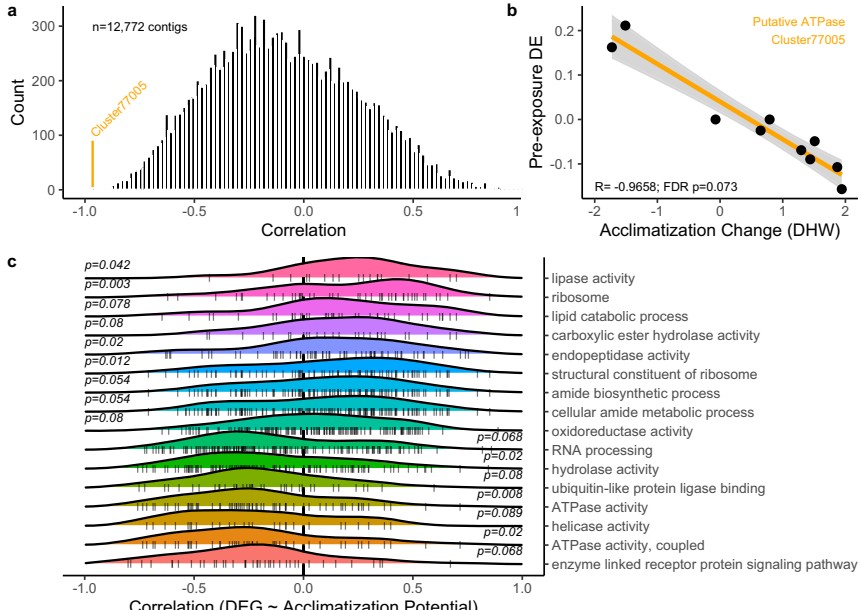

**Fig. 5 | Molecular Plasticity Explains Acclimatization Advantages. a** Each contig's ($n = 12,772$) colony specific expression change in response to constant high pre-exposure was correlated with eDHW gain or loss during stress testing. Histogram of Pearson correlation coefficients, denoting Cluster 77005, which was the only significantly correlated contig after FDR correction. **b** Correlation of Acclimatization change during stress testing with pre-exposure expression change. The orange line represents overall relationship and shaded area is 95% CI of linear regression fit. **c** Distribution of gene ontologies which were significantly enriched (FDR $p < 0.1$) in high or low correlation values. Shared x-axis with (**a**) shows background distribution. Color represents ontology. Source data are provided as a Source Data file.

moderate heat stress outside of the warmest parts of the year may influence subsequent coral bleaching, which has important implications for seasonal dynamics on coral reefs.

A significant positive change in thermal tolerance upon pre-exposure was found in two of our profiles (constant high and pulse) but was absent in the others (pulse high and pulse increase). Temperature variability can confer higher thermal tolerance on corals when compared to constant pre-exposures[42,43], but this effect is not universal[32,33,44] and our results show substantial complexity even within variable temperature regimes, although hierarchical clustering suggests that constant exposures create a discrete expression response.

It may also be possible to decouple the gene expression response between variability regimes from the ecological outcome, generating positive acclimatization responses through different mechanisms. For example, we found significant improvements in thermal tolerance during stress-testing for corals in two treatments which shared only ~15% of differentially expressed genes. Interestingly, these shared genes included functions classified as part of the 'type A' response to severe environmental stress by Dixon, Abbott[27], including innate immune response, downregulation of DNA replication and protein ubiquination. This suggests that the pulse and constant high treatment share a baseline response to thermal stress, overlaid with more specific expression responses that highlight the fine-scale sensitivity of corals[32]. This baseline response also supports the strong positive relationship between total accumulated heat stress and global changes in gene expression. The constant exposure elicited a large, unique shift in gene expression and functional differentiation that may be related to lack of relatively cooler, darker night time conditions during which corals can respond to DNA damage[54]. Using a z-score metric, we highlighted functions that distinguished the constant high treatment, finding enrichment in ribosome, peptide synthesis and metabolic function, which are hallmarks of the response of corals exposed to severe heat stress[27,55].

Strong host genotype effects on acclimatization potential underlie the signals of acclimatization found here, despite the selection of corals from a single population on a single reef for this study[50]. This sampling regime minimizes environmental variation and differences in historical selection, suggesting that neutral variation creates differing capacity for flexibility[56] which may be selected by climate-change in the future. Individual genotypes in this study demonstrated substantially and significantly different trajectories after thermal pre-exposure: up to 3.6 DHW difference (i.e., from −1.7 to +1.9 DHW) compared to controls. Genotypic differences of this magnitude demonstrate very high heritability and variance in this trait, supporting the concept of adaptive plasticity previously examined across coral populations from divergent environments[57]. Interestingly, we also note that the broad-sense heritability of acclimatization potential is elevated in *Cladocopium*-dominated corals, providing further evidence for dramatic host-derived differences based on symbiont community[23,58].

Intragenerational plasticity can be negatively[37,38] or positively[39,40] related to basal temperature tolerance, but most research on this relationship occurs at the population scale across large environmental gradients[59]. A negative relationship, which was documented in marine invertebrates under experimental evolution[60], creates a short-term limit on trait values that will impact the climate futures of certain species. Here we directly investigate this pattern within a population and do not find a negative relationship, which is of critical ecological importance on coral reefs. The *M. capitata* colonies in our study harbor different symbiont communities[51,61] which contribute strongly to historical bleaching phenotypes[50] and broadly define stress tolerance in our experiment. Despite this bimodal distribution of baseline temperature-sensitivity, there was no relationship between background resilience and ability to acquire thermal tolerance in either phenotype or the overall dataset. This result indicates that the thermally tolerant members of a coral reef community may not be limited by a 'ceiling' which is defined by their baseline trait value; if this outcome is common, it represents an important pathway for the persistence of corals. We also explored the relationship between basal and acquired thermal tolerance within phenotypes to isolate genotypic effects and

confirmed that there was no pattern. Thus, the major sources of variation in thermal tolerance in our study system, *Symbiodiniaceae* and coral genotype, do not intrinsically limit the acquisition of thermal tolerance due to gene expression plasticity.

We documented a single gene whose expression dynamics during pre-exposure explained >95% of variance in acclimatization potential 4 months later. Change in gene expression in this contig, a putative AAA + ATPase or Rho GTPase, was negatively correlated with acclimatization change such that upregulation during pre-exposure corresponded to a loss of thermal tolerance evident during stress testing. These proteins are typically involved in the energy-dependent remodeling of macromolecules in cells. Interestingly, both ATPase activity (GO:0016887) and coupled ATPase activity (GO:0042623) were almost completely disassociated with the generalized stress response in *Acropora* (*p* > 0.9), although DNA-dependent ATPase activity (GO:0008094) was significantly downregulated upon exposure to stress[27].

AAA + ATPases form a large superfamily of proteins with various functions spanning from protein unfolding and degradation through DNA recombination, replication, and repair to peroxisome biogenesis[62]. Rho GTPases serve as molecular switches, transducing extracellular signals from the cell membrane inside the cell[63], are upregulated as part of a generalized stress response[27], and are sensitive to free radical-mediated oxidation, modulating the cellular redox state[64], consistent with our previous findings that improved intracellular antioxidant defense systems enhance thermal tolerance[65]. Similarly, programmed cell-death pathways (e.g., autophagy and apoptosis) are involved in coral ability to withstand heat stress without excessive bleaching[29] and can be regulated by members of both Rho GTPase and AAA + ATPase superfamilies in model organisms[66,67].

Although the annotation of a single gene is unlikely to be causative, our global ontology enrichment analysis of these data also showed a significant negative relationship between acclimatization potential and ATPase activity and coupled ATPase activity across all 12,772 contigs analyzed. Rho GTPase function is also significantly predictive of historical bleaching phenotype in a sample of 22 *M. capitata* from Kāne'ohe Bay[50], a subset of which were used for this study, supporting a pathway by which differential gene expression during thermal pre-exposure impacts subsequent phenotype.

The results presented here indicate that durable phenotypic plasticity can be induced by short-term thermal pre-exposure in corals. We also document substantial variation in acclimatization potential among genotypes from the same source reef, representing a cryptic source of heritable differentiation on which selection can act. This variation is unrelated to basal thermal tolerance, suggesting that adaptive plasticity, symbiont community and fixed differences between genotypes can contribute to the long-term persistence of coral reefs under climate change.

## Methods

### Experimental design and pre-exposure
Ten known genotypes of *Montipora capitata*[50] were collected from Reef 13 in Kāne'ohe Bay in May 2019 (DAR permit SAP-2020-25 to HIMB), equally split between each of two historical bleaching phenotypes originally tagged in the 2015 bleaching event[68,69]. These samples were equally split between *Cladocopium*- and *Durusdinium*-dominated colonies[50,51]. Fragments (*n* = 500) were returned to the Hawai'i Institute of Marine Biology, mounted on labeled plugs, and transferred to indoor mesocosms (Fig. 1). After 2 weeks of indoor acclimation, corals were exposed to either a control or one of four short-term heat pre-exposure profiles (constant high, pulse, pulse increase, pulse high) between 28 °C and 31.5 °C for 5 days (Fig. 1b) in replicate tanks (*n* = 2). These treatments accumulated 1.1–2.7 eDHW (Fig. 1c) had a -9-fold range in rate of change (°C/hr; Fig. 1d). After this pre-exposure, a subset of corals (*n* = 5–7 fragments per colony per treatment, *N* = 299) were returned to the

collection reef on August 15th and mounted on a platform at -2 m depth for recovery prior to subsequent stress testing 124 days later.

### RNA sequencing
At noon on the day before the start of pre-exposure profiles, we collected RNA samples with a dermal curette (-2 mm core) from a subset of 50 designated fragments, immediately placing them in liquid nitrogen and storing at −80 °C until processing. We repeated this sampling at noon on day 5 of the pre-exposure (Fig. 1b) when all treatments (excluding control) were at 31.5 °C. At each timepoint we collected one sample from each fragment, totaling 5 samples per phenotype (*n* = 2) per treatment (*n* = 5) per timepoint (*n* = 2), for a total of 100 samples. All colonies were sampled using the same minimally-invasive methodology, and while we cannot exclude the potential for sampling stress, the relative stress for each treatment and individual coral should be equivalent. We used a Trizol RNA Extraction with DNase treatment followed by phenol-chloroform purification and prepared 3' Tag-Seq libraries following[24]. Libraries were randomly allocated to two lanes and sequenced (SR 100 bp) at UC Davis on a HiSeq 4000. Raw reads were de-duplicated using tagseq_clipper.pl (https://github.com/z0on) and trimmed with *Cutadapt 3.7*[70] to remove poly-A tails and low quality bases. We quantified filtered reads using Salmon 1.8[71] (--validateMappings, --gcBias, --seqBias) against a transcriptome prepared from[72]. To prepare the reference and isolate host sequences, we used *bwa-mem* 0.7.17[73] to align the metatranscriptome against the *M. capitata* genome[74] and excluded unmapped contigs before clustering similar sequences using cd-hit-est 4.8.1 (-c 0.95 -G 0 -aL 0.3 -aS 0.3)[75]. We used *blastx* 2.12.0 against the Uniprot/Swissport KB to annotate contigs and retrieve gene ontologies, using the best hit with *e* value < 10⁻⁶. We manually annotated genes of interest using *blast*, *MotifFinder* (https://www.genome.jp/tools/motif/), *OrthoDB* 10.1[76], *Interpro*[77], *Pfam*[78], and hidden Markov models.

### Stress testing
We retrieved corals from the field in November 2019 (124 days after the conclusion of the pre-exposures) and randomly allocated them into two tanks for stress testing. Each tank was heated from 28 °C to 32.5 °C over 10 days and then maintained at 32.5 °C (Supplemental Fig. 1). We collected photosynthetic efficiency data from each fragment (*fv/fm*; dark-adapted quantum yield; Walz Diving PAM and WalzWinControl 3) in duplicate before the temperature ramp to assess latent impacts of pre-exposure and confirm that photochemical data had a consistent baseline. We re-collected PAM timepoints every 2–3 days during the stress test (Supplemental Fig. 1). Additional detail on corals used in this experiment and their symbiont dynamics can be found in Dilworth, Caruso[51]. See Fig. 1a for details of the experimental timeline.

### Analysis
All data analysis was conducted in R 4.1.2. We used 28.5 °C (mean monthly maximum +1 °C)[51] as the threshold to calculate Degree Heating Weeks (DHW), a measure of the total accumulation of heat stress above the typical warmest conditions for a location which is integrative of temperature and duration. We used this value to calculate experimental DHW (eDHW)[79] for pre-exposure. We compared raw *fv/fm* before stress testing across phenotypes and treatments using a two-way ANOVA to ensure recovery from pre-exposure treatments. We calculated *fv/fm* relative to averaged initial *fv/fm* for each fragment in the stress test before fitting dose response curves for each treatment and phenotype combination using a three parameter Weibull Model (W1.3) with the upper limit fixed at 1 in *drc* 3.0[80]. We calculated accumulated DHW until ED10 (10% decline in relative *fv/fm*) as an arbitrary threshold because it maximized differentiation between treatments and encompassed the fitted range of all analyses. The difference between ED10 values of different treatments represents the gain or loss of thermal tolerance (in DHW) for each comparison of interest,

which we term acclimatization potential. We used the estimated mean and standard error of ED10 from *drc* to calculate t-statistics and *p* values comparing the control to each heated treatment for each phenotype independently.

We subset data to include only control and constant high treatments and re-fit dose response curves for each colony, excluding observations with Cook's distance >3× the mean. We used the estimated mean and standard error of ED10 from *drc* to calculate t-statistics comparing control and constant high treatments for each colony. We used a bootstrap approach ($n = 1000$ replicates per colony × treatment) to calculate acclimatization potential for each genotype using the mean and standard errors from the *drc* output. These values were analyzed with a one-way ANOVA to calculate variance explained by colony, which was divided by the total variance to calculate broad-sense heritability.

We imported raw reads from Salmon and filtered to contigs averaging at least 5 reads per sample to create a dataset of 17,826 host-derived contigs. We used a variance stabilizing transformation in *DESeq2* 1.34[81] to normalize data for visualization and examined global patterns using PERMANOVAs in *vegan* 2.5[82] with Manhattan distances. After examining these patterns, we analyzed differential expression in *DEseq2* between timepoints for each treatment, accounting for the phenotype, colony, and the concurrent change in the control treatment (e.g., contrast = timepoint, timepoint2.treatment). We compared mean $\log_2$ fold-change between treatments using a one-way ANOVA and used a linear regression to compare mean $\log_2$ fold-change with eDHW and rate of change. We examined overlapping significance between treatments using venn diagrams. After observing the distinctive response in the constant high pre-exposure, we calculated a *z*-score for each contig comparing constant high to the pulse, pulse high, and pulse increase treatments to examine functional responses unique to the constant high treatment. We used these *z*-scores as heats for GO_MWU[83], a rank-based ontology enrichment approach that defines functions significantly enriched in large values of a continuous variable. We examined overlap in significant ontologies and those responding to the generalized stress response in a metanalysis of *Acropora*[27] using ranks and *p* values from a GO_MWU analysis comparing gene expression between controls and six different stressors.

To evaluate the relationship between gene expression change and acclimatization potential (see above), we used the variance stabilized data to independently calculate the constant high-specific response (control-corrected) for each colony and performed a Pearson correlation between each contig and the acclimatization potential. We used the rho values from this correlation as heats for GO_MWU to examine enrichment in functions relating to acclimatization potential.

### Reporting summary
Further information on research design is available in the Nature Research Reporting Summary linked to this article.

## Data availability
The raw sequencing data generated in this study is available at NCBI under Bioproject PRJNA847955. This study used existing data from the pfam database (hosted at pfam.xfam.org), the InterPro database (hosted at ebi.ac.uk/Interpro), and the OrthoDB database (hosted at orthodb.org). Source data are provided with this paper, including data stress-testing response, transcriptomic response, heritability calculations and experimental conditions. Source data are provided with this paper.

## Code availability
All processed sequencing data and analytical code is available at github.com/druryc/mac_19 or under https://doi.org/10.5281/zenodo.6877825[84].

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

## Acknowledgements
We are grateful to Amanda Williams for providing a reference meta-transcriptome for this work. We would like to thank the Nina Bean, Luke Kikukawa, Valerie Kahkejian, Mariana Rocha de Souza, the Coral Resilience Lab for support. We also thank John Talabot, Andrew Weatherall and Matthew Dear for analytical support. This project was funded by the Paul G. Allen Family Foundation. J.G. was supported by the U.S. Geological Survey's (USGS) Environmental Health Program (Contaminant Biology and Substances Hydrology). Any use of trade, firm, or product names is for descriptive purposes only and does not imply endorsement by the U.S. Government. Corals were collected under DAR permit# SAP-2020-25 to HIMB. We thank Ruth Gates, who encouraged us to think creatively about solutions for coral reefs facing climate change. This is SOEST contribution number 11549 and HIMB contribution 1897.

## Author contributions
C.D., J.D., C.C., E.M. conceived the project. C.D., J.D., E.M., J.G. analyzed data. All authors collected data and wrote the paper.

## Competing interests
The authors declare no competing interests.
