## [Peer Review File · Nature Communications]

Expression plasticity regulates intraspecific variation in the acclimatization potential of a reef-building coralReviewers' Comments:

Reviewer #1:

Remarks to the Author:

This is a very interesting paper, super relevant for both theoretical and practical coral biology and really carefully done. The ideas, experiments, and analyses are highly original and because of that may be confusing for some readers, but the authors do a really good job explaining it all (with a few minor exceptions, see below). I only have relatively minor suggestions and optional alternative analyses that in all probability will not change the main results. Below I list my comments in the approximate order from the most to least important.

Introduction: Instead of the general, broadly known and agreed upon, things about coral bleaching and possible role of plasticity in adaptation (yawn), I would be much more interested in seeing an overview of research on coral stress hardening, and any controversies that exist there (right now much of it is in the discussion - please move it up). I feel that would make the paper a more exciting read.

L291: "Here we show that shifts in gene expression during pre-exposure create a latent, durable improvement in thermal tolerance in a reef-building coral." - I agree that this is a likely explanation of the results observed, but still, there is no formal ground to claim the causality - that coral gene expression change actually causes changes in stress tolerance. Maybe it is some third factor (e.g. symbionts) that influences both. Please change "create" to "are associated with".

L321 and L372-374: How are these responses related to the generalized stress response in *Acropora* (Dixon et al 2020, DOI: 10.1111/mec.15535)? I understand it will be very hard to compare on a gene by gene basis because these datasets cannot be mapped to the same reference, but one can easily compare the functional responses (GO_MWU or KOGMWU delta-ranks).

L164: Ordination of log₂-transformed gene expression data (which is what the vst function in DESeq2 outputs): I am not sure what justifies the use of Bray-Curtis distances here. BC is a non-metric distance for species abundance (counts) data, it doesn't make much sense for log-transformed gene expression. It is more appropriate to use Manhattan distance divided by the number of genes - in which case the distance between two samples is the average log-fold change (across all genes) between them.

L332: broad-sense heritability of the acclimatization potential is simply the proportion of its variation explained by genotype (after controlling for the effect of dominant symbiont, I would suggest); I don't see why not calculate it here?

Gene expression analysis is quite complicated but I think I got it: The authors first computed gene expression changes between timepoints for each treatment (in DESeq2), and then compared these changes among treatments, using as reference the constant high one (using ANOVA, for each contig). This is statistically convoluted but makes sense. Yet, this is the analysis strongly driven by experimental design expectations, ie strongly "supervised". In complicated experimental scenarios like this one I would be really interested in looking at an unsupervised analysis by WGCNA (maybe after removing the covariates using `limma::removeBatchEffect` function) - it may reveal interesting things the authors might have overlooked while pursuing their design scheme. I am 90% sure the authors have actually tried WGCNA, so how did it go? (if you did not do WGCNA, feel free to ignore this).

L166: How exactly did you set up the DESeq2 model to account for these things? Especially "the concurrent change in the control treatment", I am having difficulty visualizing that - is it treatment:timepoint interaction term you are after?

Is the "phenotype" here - historically bleached vs historically non-bleached - the same as dominant

symbiont (C vs D)? If yes, it might make it easier for readers if "phenotype" is replaced by "dominant symbiont" in figures and text ("phenotype" in my mind is response to treatment, which is not the case here, so I am getting a bit confused). Another reason is that the type of symbiont is likely to be the actual driver of observed differences between these two groups.

L203: "Shared axis with panel A." looks out of place

Fig. 4d: can you please also cluster the rows? (function pheatmap::pheatmap) Might look better.

Also Fig 4d: Please try a color scheme with a neutral color (gray or white) for the middle of the scale (zero change).

Fig. 4B the venn diagram is not really helpful. Is it possible to present the message we are supposed to see here in some other graphical form, more straightforwardly? I think fig 4D actually does this (the clustering tree on top), so maybe move the venn to supplements?

Figures: Viridis color scheme looks nice for sure but does not allow for easy visual discrimination between categorical groups (for example pulse high and constant high are visually indistinguishable as colored lines). In addition, while viridis is very appropriate for quantitative variables or ordered categories, here there seems to be no a priori reason to assume progression of treatments in the chosen viridis-order (control, pulse increase, pulse, pulse high, constant high), so maybe it is more appropriate to use some "diverging" color scheme?

Please list FDR values for the GO categories on Fig. 5 C (cool graph by the way!)

L265: Cluster 77005 story: how was the FDR of 0.073 determined? From the graph on Fig. 5A it looks like this gene is quite likely just the tail of the overall distribution... Is it worth so much attention?

Reviewer #2:

Remarks to the Author:

In the present manuscript, the authors take advantage of the previous characterization of bleaching phenotypes in *Montipora capitata* corals to ascertain the contribution of genotype, symbiont communities and previous history of heat exposure to coral plasticity and resistance to thermal stress. For that purpose, the present work used 10 genotypes equally distributed across the 2 bleaching phenotypes. The results revealed higher thermal stress resistance in particular genotypes subject to particular heating treatments and independently from bleaching phenotype. The authors conclude that durable phenotypic plasticity can be induced by short-term thermal pre-exposure in corals and that substantial variation in acclimatization potential is provided by particular genotypes, along with adaptive plasticity, symbiont community and other fixed differences.

Overall, I found the present study engaging and interesting, particularly by leveraging the previous bleaching phenotype data from *Montipora* in Kane'ohe Bay. The combination of that with different experimental heating treatments adds an additional layer of interest as to how the heating regime can potentially alter acclimatization. The data analyses and results seem solid, however, there are a few comments I'd like to make which I hope help strengthen the manuscript.

Major comments

Line 95. Samples seem to be equally distributed between *Caldocopium* and *Durusdinium*, however, the details about this are lacking throughout the manuscript I think. I missed any reference to quantitative analyses of symbionts (amounts, have they changed?) on the different genotypes or phenotypes

during the study, although the symbiont differences are cited as one of the potential causes contributing to the differences in thermal performance observed.

Line 110. Based on the sampling methodology cited, coral colonies were recurrently sampled throughout the study at each time point. I would like to ask authors to address how do they ensure, gauge, or control that the recurrent stress caused by the sampling of the same colony does not impact the baseline stress and subsequent gene expression patterns observed in the corals, and if so, how is this effect controlled in data analyses.

Line 183. Does that mean that it took 124 days for both phenotypes to show an absence in photosynthesis efficiency between them? How was this measured? Periodically? Please explain.

Line 317. About the shared 18% differentially genes in thermal responses, it would be interesting to ascertain if those genes are involved in basal stress responses that can be complemented by additional genes (different in different treatments) to produce the same thermal performance phenotype effect.

Line 356. Is it a single gene or a single contig on which several genes (including the differentially expressed) are included?

Line 367. I am personally reluctant to cite biorxiv works as peer reviewed references. I don't know if citation 75 has been published, if not, I would suggest adding something like "unpublished data".

Minor comments

Lines 52-64. In this section, the authors summarize the coral responses and underlying mechanisms including gene expression variation. I feel this paragraph will be strengthened by adding references for epigenetic regulation of gene expression, several papers have been published on the issue, notably Eirin-Lopez and Putnam *Annu Rev Mar Sci*. In addition, a recent report by Eirin-Lopez's lab links differential symbiont communities with heterogeneous epigenetic responses (Rodriguez-Casariago et al. *Mol Ecol* 2022 31:588)

Line 101. Please indicate the length of recovery time before subsequent stress testing.

Line 143. I feel that, for the non-expert audience, the concept of Degree Heating Week should be described.

Update reference 41.

Reference 44 lacks details of citation.

Reviewer #1 (Remarks to the Author):

This is a very interesting paper, super relevant for both theoretical and practical coral biology and really carefully done. The ideas, experiments, and analyses are highly original and because of that may be confusing for some readers, but the authors do a really good job explaining it all (with a few minor exceptions, see below). I only have relatively minor suggestions and optional alternative analyses that in all probability will not change the main results. Below I list my comments in the approximate order from the most to least important.

Thank you very much for the kind words and specific advice for improving the work.

Introduction: Instead of the general, broadly known and agreed upon, things about coral bleaching and possible role of plasticity in adaptation (yawn), I would be much more interested in seeing an overview of research on coral stress hardening, and any controversies that exist there (right now much of it is in the discussion - please move it up). I feel that would make the paper a more exciting read.

We have added this context to the introduction and agree that it is appropriate, however this is more of an addition than a replacement and we have elected to maintain much of the previous text as it is important context for non-specialists in a more general journal. This shifted text has also improved the flow of the discussion and we appreciate the suggestion.

L291: “Here we show that shifts in gene expression during pre-exposure create a latent, durable improvement in thermal tolerance in a reef-building coral.” - I agree that this is a likely explanation of the results observed, but still, there is no formal ground to claim the causality - that coral gene expression change actually causes changes in stress tolerance. Maybe it is some third factor (e.g. symbionts) that influences both. Please change “create” to “are associated with”.

Thank you for this suggestion, amended.

L321 and L372-374: How are these responses related to the generalized stress response in *Acropora* (Dixon et al 2020, DOI: 10.1111/mec.15535)? I understand it will be very hard to compare on a gene by gene basis because these datasets cannot be mapped to the same reference, but one can easily compare the functional responses (GO_MWU or KOGMWU delta-ranks).

This is a great suggestion and we appreciate it. We've been able to extract some information, but the github for this project is difficult to navigate (hundreds upon hundreds of files), making a more thorough association of data between projects difficult. However, we were able to extract some cool patterns, which we have discussed.

“Among these ontologies, 8 of 11 were also significantly upregulated in a meta-analysis of the generalized stress response in Acropora [1], including peptide metabolism and biosynthesis, cellular responses to endogenous stimulus and structural constituents of ribosomes.”

“Rho GTPases serve as molecular switches, transducing extracellular signals from the cell membrane inside the cell [2], are upregulated as part of a generalized stress response [1], and are sensitive to free radical-mediated oxidation, modulating the cellular redox state [3], consistent with our previous findings that improved intracellular antioxidant defense systems enhance thermal tolerance [4].”

“Interestingly, both ATPase activity (GO:0016887) and coupled ATPase activity (GO:0042623) were almost completely disassociated with the generalized stress response in Acropora ($p > 0.9$), although DNA-dependent ATPase activity (GO:0008094) was significantly downregulated upon exposure to stress [1]. This contrast suggests a background function for ATPase in coral stress management which is fixed and may not actively respond to thermal stress.”

“Despite the distinct molecular responses of each pre-exposure, both the constant high and pulse treatments elicited a shared significant response in 15% of genes that were significant in at least one treatment. These genes, which may represent core molecular responses to thermal challenge, included many of the ontologies in a generalized response to severe stress [1], including the activation of an innate immune response, positive regulation of protein ubiquitination and apoptosis and downregulation of replication.”

L164: Ordination of log₂-transformed gene expression data (which is what the vst function in DESeq2 outputs): I am not sure what justifies the use of Bray-Curtis distances here. BC is a non-metric distance for species abundance (counts) data, it doesn't make much sense for log-transformed gene expression. It is more appropriate to use Manhattan distance divided by the number of genes - in which case the distance between two samples is the average log-fold change (across all genes) between them.

We appreciate this point and attention to detail. We have amended this part of the analysis, which did not change any significance and barely adjusted p-values (although f-values are marginally different). The text and figures have been amended.

L332: broad-sense heritability of the acclimatization potential is simply the proportion of its variation explained by genotype (after controlling for the effect of dominant symbiont, I would suggest); I don't see why not calculate it here?

We had previously used the dose response mean and standard errors to calculate p-values from a t-test comparing each treatment within each genotype and can similarly bootstrap broad-sense heritability, which we find is high: ~0.89. See details below for why these calculations cannot be completed on raw data.

We have added details on this approach to the methods and results and added a figure.

“We used a bootstrap approach (n=1000 replicates per colony x treatment) to calculate acclimatization potential for each genotype using the mean and standard errors from the drc output. These values were analyzed with a one-way ANOVA to calculate variance explained by colony, which was divided by the total variance to calculate broad-sense heritability.”

“Bootstrapped estimates of broad-sense heritability (H^2) of acclimatization potential was 0.897 for all corals (Figure 3a) and acclimatization potential was significantly different between genotypes (one-way ANOVA, $p < 0.001$). Broad-sense heritability estimates were nearly 35% higher when analyzing only historically bleached corals ($H^2 = 0.918$) than when analyzing only historically nonbleached corals ($H^2 = 0.681$). There was little explanatory power and no significant relationship between basal thermal tolerance and acclimatization potential (Figure 3; $R^2 = 0.187$, $p = 0.212$). This relationship was not significant when examining only bleached ($p = 0.192$) or nonbleached corals ($p = 0.335$) and was not significant when removing two large negative outliers (Figure 3b; $p = 0.334$).”

We originally did not calculate this value because the experimental design prevents us from doing so. In short, the ED10 and acclimatization potential were calculated on curve fits for ‘groups’ of samples representing all fragments in a given treatment x genotype combination. This approach, which leverages the 6-7 fragments available, minimizes the noise from individual outlier fv/fm observations, but at the expense of replication. That is, there were not ‘replicate’ ed10 values within each treatment x genotype combination due to having multiple fragments.

Gene expression analysis is quite complicated but I think I got it: The authors first computed gene expression changes between timepoints for each treatment (in DESeq2), and then compared these changes among treatments, using as reference the constant high one (using ANOVA, for each contig). This is statistically convoluted but makes sense. Yet, this is the analysis strongly driven by experimental design expectations, ie strongly “supervised”. In complicated experimental scenarios like this one I would be really interested in looking at an unsupervised analysis by WGCNA (maybe after removing the covariates using `limma::removeBatchEffect` function) - it may reveal interesting things the authors might have overlooked while pursuing their design scheme. I am 90% sure the authors have actually tried WGCNA, so how did it go? (if you did not do WGCNA, feel free to ignore this).

We did pursue WGCNA in the initial analyses of this manuscript, but ultimately favored the more specific presented here because it maximizes information about genotype nested within phenotype. The primary interesting outcomes from this approach were not coordinated responses within or between treatments, but rather baseline differences between phenotypes, which we thought was appropriately captured by the PCA in figure 4a.

Its possible to include more WGCNA information, but we are concerned that it will make the manuscript a bit unwieldy especially given how much ecological and experimental data we have. Since the reviewer doesn’t seem to indicate that this is make or break, we have elected not to include the WGCNA analysis, but if this is an obstacle we can find a solution.

L166: How exactly did you set up the DESeq2 model to account for these things? Especially “the

concurrent change in the control treatment”, I am having difficulty visualizing that - is it treatment:timepoint interaction term you are after?

Thanks for your attention to detail. We'd like to note that the code is now on github for those who are really interested in the nitty-gritty. The overall model was set up with timepoint, treatment, phenotype (see F4a for huge differences between phenotype, which needed to be accounted for), colony nested within phenotype and treatment nested within timepoint. We used control and T1 as the reference levels. You are correct that the treatment:timepoint interaction is part of this, alongside the change through time in the control. We extracted the contrasts comparing timepoint (e.g., timepoint_T2_vs_T1) and treatment at T2 (e.g., timepointT2.treatmentconstant_high). This design extracts differentially expressed genes at T2 between a given treatment and the control, while accounting for the change over time in the control to properly extract relative expression levels specific to the treatment.

Is the “phenotype” here - historically bleached vs historically non-bleached - the same as dominant symbiont (C vs D)? If yes, it might make it easier for readers if “phenotype” is replaced by “dominant symbiont” in figures and text (“phenotype” in my mind is response to treatment, which is not the case here, so I am getting a bit confused). Another reason is that the type of symbiont is likely to be the actual driver of observed differences between these two groups.

This is a good question and aligns with some of R2s comments about the symbionts. In short, we have maintained the non/bleached terminology here because A) it is strongly but not completely driven by symbionts [5, 6], with a substantial host influence [7] and B) to maintain consistency with the other published projects that examine this cohort of corals [8]. As an aside, published [9] and unpublished data from our lab shows there is a substantial minority of ‘nonbleached’ corals that harbor Cladocopium and ‘bleached’ corals that harbor ‘Durusdinium’, so this relationship is not as tight as the samples within the present manuscript make it appear.

We have added details in the methods, background and results to better situate this information and hope the reviewer finds it acceptable.

L203: “Shared axis with panel A.” looks out of place

Thank you for pointing this out, this figure has been revised this line removed.

Fig. 4d: can you please also cluster the rows? (function pheatmap::pheatmap) Might look better. Also Fig 4d: Please try a color scheme with a neutral color (gray or white) for the middle of the scale (zero change).

We've changed both of these for clarity in this figure, thank you for the suggestion.

Fig. 4B the venn diagram is not really helpful. Is it possible to present the message we are

supposed to see here in some other graphical form, more straightforwardly? I think fig 4D actually does this (the clustering tree on top), so maybe move the venn to supplements?

Agreed, we have moved it to supplemental materials alongside some additional files.

Figures: Viridis color scheme looks nice for sure but does not allow for easy visual discrimination between categorical groups (for example pulse high and constant high are visually indistinguishable as colored lines). In addition, while viridis is very appropriate for quantitative variables or ordered categories, here there seems to be no a priori reason to assume progression of treatments in the chosen viridis-order (control, pulse increase, pulse, pulse high, constant high), so maybe it is more appropriate to use some “diverging” color scheme?

There doesn't seem to be a strong consensus on the use of viridis in this context and we would point out that the treatments are displayed as ordered categories according to overall accumulated heat stress (eDHW in the paper), so this is not a random choice of unrelated categorical factor levels. Given that viridis is also considered to be favorable for colorblindness and the reviewer didn't indicate this is make or break, we have elected to maintain the color scheme.

Please list FDR values for the GO categories on Fig. 5 C (cool graph by the way!)

Added, thanks for the suggestion and positive feedback.

L265: Cluster 77005 story: how was the FDR of 0.073 determined? From the graph on Fig. 5A it looks like this gene is quite likely just the tail of the overall distribution... Is it worth so much attention?

First, p-values were determined from the correlation between each individual gene's expression and relative acclimatization response using `Hmisc::rcorr`. We then used the `p.adjust` function `stats::p.adjust` with `method="fdr"` on these values.

As for your question about the importance of Cluster77005, it's a fair point. It is actually somewhat discrete from the distribution, but is not a clear outlier. However, the importance of these functional annotations in the GO_MWU analysis, which produces the same strongly significant negative relationship for ATPase activity overall suggests that this is an important pattern. Regardless, we appreciate the point and have adjusted the text to provide better context.

Reviewer #2 (Remarks to the Author):

In the present manuscript, the authors take advantage of the previous characterization of bleaching phenotypes in *Montipora capitata* corals to ascertain the contribution of genotype, symbiont communities and previous history of heat exposure to coral plasticity and resistance to thermal stress. For that purpose, the present work used 10 genotypes equally distributed across the 2 bleaching phenotypes. The results revealed higher thermal stress resistance in particular genotypes subject to particular heating treatments and independently from bleaching phenotype. The authors conclude that durable phenotypic plasticity can be induced by short-term thermal pre-exposure in corals and that substantial variation in acclimatization potential is provided by particular genotypes, along with adaptive plasticity, symbiont community and other fixed differences.

Overall, I found the present study engaging and interesting, particularly by leveraging the previous bleaching phenotype data from *Montipora* in Kane'ōhe Bay. The combination of that with different experimental heating treatments adds an additional layer of interest as to how the heating regime can potentially alter acclimatization. The data analyses and results seem solid, however, there are a few comments I'd like to make which I hope help strengthen the manuscript.

Thank you for the constructive criticisms and positive feedback, we appreciate your suggestions.

Major comments

Line 95. Samples seem to be equally distributed between *Cladocopium* and *Durusdinium*, however, the details about this are lacking throughout the manuscript I think. I missed any reference to quantitative analyses of symbionts (amounts, have they changed?) on the different genotypes or phenotypes during the study, although the symbiont differences are cited as one of the potential causes contributing to the differences in thermal performance observed.

*Thank you for this point, R1 mentions a similar issue and we agree that the context for this information was not properly presented. We have revised the methods and results to point readers (and the reviewer) toward a separate manuscript that covers these issues in great detail [5]. As far as the distribution mentioned above, this is correct, samples were equally split between *Cladocopium*- and *Durusdinium*-dominated colonies, and we have updated the methods section to clarify that these differences in symbiont community composition correspond to the historical bleaching phenotypes in this study. Previous work [5-7] has shown that the dominant symbiont genus that these colonies associate with is very stable and that the dominant symbiont type in each bleaching phenotype persisted throughout multiple thermal stress events. We have added this clarifying information to the results section as well.*

*Lastly, phenotype in this system is strongly but not completely driven by symbionts [5, 6], with a substantial host influence [7]. As an aside, published [9] and unpublished data from our lab shows there is a substantial minority of ‘nonbleached’ corals that harbor *Cladocopium* and ‘bleached’ corals that harbor *Durusdinium*, so this relationship is not as tight as the samples within the present manuscript make it appear.*

Line 110. Based on the sampling methodology cited, coral colonies were recurrently sampled throughout the study at each time point. I would like to ask authors to address how do they ensure, gauge, or control that the recurrent stress caused by the sampling of the same colony does not impact the baseline stress and subsequent gene expression patterns observed in the corals, and if so, how is this effect controlled in data analyses.

This is an interesting point and we appreciate the reviewer’s bringing it to our attention. Ultimately, we cannot control for the stress of sampling except for noting that it was deliberately equal across all treatments and colonies and that all corals were sampled in the same way. This should, conceptually, mean that treatments are still properly controlled and the only differences are between pre-exposures. We’d also note that our DEseq approach accounts for change over time between T1 and T2 in the control treatment, so none of the differential expressed genes we examine downstream should be impacted by sampling stress (i.e., they would have been DE in the entire dataset, controls inclusive, and thereby not documented as robust changes based on our model setup).

“All colonies were sampled using the same minimally-invasive methodology, and while we cannot exclude the potential for sampling stress, the relative stress for each treatment and individual coral should be equivalent.”

Line 183. Does that mean that it took 124 days for both phenotypes to show an absence in photosynthesis efficiency between them? How was this measured? Periodically? Please explain.

No, the initiation of the stress-test was not predicated on any information about the ‘recovery’ after pre-exposure, we simply sought to confirm that corals were entering the stress from the same starting point. Dilworth, Caruso [5] has detail about PAM values in the time between pre-exposure and stress testing, including a timepoint in October (~6 weeks before stress test) where fragments from the same experiment did show a small but significant difference between phenotypes and treatments. We were simply confirming that this difference was no longer apparent in the present work. We have added detail to the results and methods.

“We retrieved corals from the field in November 2019 (124 days after the conclusion of the pre-exposures) and randomly allocated them into two tanks for stress testing. Each tank was heated from 28°C to 32.5°C over 10 days and then maintained at 32.5°C (Supplemental Figure 2). We collected photosynthetic efficiency data from each fragment (fv/fm; dark-adapted quantum yield; Walz Diving PAM) in duplicate before the temperature ramp to assess latent impacts of pre-exposure and confirm that photochemical data had a consistent baseline. We re-collected pam timepoints every 2-3 days during the stress test (Supplemental Figure 2).

Additional detail on corals used in this experiment and their symbiont dynamics can be found in Dilworth, Caruso [5]. See Figure 1a for details of the experimental timeline.”

Line 317. About the shared 18% differentially genes in thermal responses, it would be interesting to ascertain if those genes are involved in basal stress responses that can be complemented by additional genes (different in different treatments) to produce the same thermal performance phenotype effect.

Great idea, thank you for the suggestion. We ultimately used R1s suggestion of making comparisons with Dixon et al. 2020’s meta-analysis of the generalized stress response of corals. This comparison allowed us to show that many of the baseline canonical severe stress response genes are found in this overlapping segment of our data – unfolded protein response, protein ubiquitination, activation of the innate immune response and downregulation of DNA replication. Than you again for the idea, this has definitely enriched the context and discussion in this paper.

“Despite the distinct molecular responses of each pre-exposure, both the constant high and pulse treatments elicited a shared significant response in 15% of genes that were significant in at least one treatment. These genes, which may represent core molecular responses to thermal challenge, included many of the ontologies in a generalized response to severe stress [1], including the activation of an innate immune response, positive regulation of protein ubiquitination and apoptosis and downregulation of replication.”

Line 356. Is it a single gene or a single contig on which several genes (including the differentially expressed) are included?

This is a single gene, contigs were derived from individual mRNAs (this one is ~800bp).

Line 367. I am personally reluctant to cite biorxiv works as peer reviewed references. I don’t know if citation 75 has been published, if not, I would suggest adding something like “unpublished data”.

According to the editor’s recommendation this should be cited as a preprint and not as unpublished data, so we have maintained this citation.

Minor comments

Lines 52-64. In this section, the authors summarize the coral responses and underlying mechanisms including gene expression variation. I feel this paragraph will be strengthened by adding references for epigenetic regulation of gene expression, several papers have been published on the issue, notably Eirin-Lopez and Putnam Annu Rev Mar Sci. In addition, a recent report by Eirin-Lopez’s lab links differential symbiont communities with heterogeneous epigenetic responses (Rodriguez-Casariago et al. Mol Ecol 2022 31:588)

Thank you for pointing this out, it was certainly an oversight on our part. We have included these references and a few others throughout the introduction and discussion.

Line 101. Please indicate the length of recovery time before subsequent stress testing.

Amended, methods now at the end of the manuscript: "After this pre-exposure, a subset of corals (n=5-7 fragments per colony per treatment, N=299) were returned to the collection reef on August 15th and mounted on a platform at ~2m depth for recovery prior to subsequent stress testing 124 days later."

Line 143. I feel that, for the non-expert audience, the concept of Degree Heating Week should be described.

Thanks for this suggestion, updated: "All data analysis was conducted in R. We used 28.5°C (mean monthly maximum +1°C)[5] as the threshold to calculate Degree Heating Weeks (DHW), a measure of the total accumulation of heat stress above the typical warmest conditions for a location which is integrative of temperature and duration. We used this value to calculate experimental DHW (eDHW) [10] for pre-exposure and stress testing treatments."

Update reference 41.

Updated, thanks for noticing :)

Reference 44 lacks details of citation.

Thank you for pointing this out, we have added this information.

References

1. Dixon, G., E. Abbott, and M. Matz, *Meta-analysis of the coral environmental stress response: Acropora corals show opposing responses depending on stress intensity*. *Molecular Ecology*, 2020. **29**(15): p. 2855-2870.
2. Moon, S.Y. and Y. Zheng, *Rho GTPase-activating proteins in cell regulation*. *Trends in cell biology*, 2003. **13**(1): p. 13-22.
3. Hobbs, G.A., et al., *Rho GTPases, oxidation, and cell redox control*. *Small GTPases*, 2014. **5**(2): p. e28579.
4. Majerová, E. and C. Drury, *A BI-1 mediated cascade improves redox homeostasis during thermal stress and prevents oxidative damage in a preconditioned reef-building coral*. *bioRxiv*, 2021.
5. Dilworth, J., et al., *Host genotype and stable differences in algal symbiont communities explain patterns of thermal stress response of Montipora capitata following thermal pre-exposure and across multiple bleaching events*. *Coral Reefs*, 2020.
6. Drury, C., et al., *Intrapopulation adaptive variance supports thermal tolerance in a reef-building coral*. *Communications Biology*, 2022. **5**(1): p. 1-10.

7. Roach, T.N., et al., *Metabolomic signatures of coral bleaching history*. Nature Ecology & Evolution, 2021: p. 1-9.
8. Drury, C., et al., *Ecosystem-scale mapping of coral species and thermal tolerance*. Frontiers in Ecology and the Environment, 2022.
9. Cunning, R., R. Ritson-Williams, and R.D. Gates, *Patterns of bleaching and recovery of Montipora capitata in Kāneʻohe Bay, Hawaiʻi, USA*. Marine Ecology Progress Series, 2016. **551**: p. 131-139.
10. Leggat, W., et al., *Experiment Degree Heating Week (eDHW) as a novel metric to reconcile and validate past and future global coral bleaching studies*. Journal of Environmental Management, 2022. **301**: p. 113919.

Reviewers' Comments:

Reviewer #1:

Remarks to the Author: I'm quite satisfied with the authors' response, I am even warming up to the idea that ATPase genes might indeed be related to acclimatization potential :) Great job! Let's publish this

Reviewer #2:

Remarks to the Author:

Thanks to authors for the careful work addressing all comments. I recommend the present manuscript for publication in its present form.